# Gradient Heatmetry and PIV Investigation of Heat Transfer and Flow near Circular Cylinders

**Vladimir Seroshtanov * and Andrey Gusakov**

Science Educational Center "Energy Thermophysics", Peter the Great St. Petersburg Polytechnic University (SPbPU), St. Petersburg 195251, Russia
* Correspondence: serosht_vv@spbstu.ru; Tel.: +7-812-552-7773

**Abstract:** This paper presents an experimental investigation of convective heat transfer around circular cylinders installed one after another. The experimental approach based on the combined application of gradient heatmetry and particle image velocimetry. The method made it possible to measure velocity fields and heat flux simultaneously. Investigation of the flow characteristics and the heat transfer coefficient distribution over a system of cylinders was carried out for various Reynolds numbers in the range of Re = $(4...40) \times 10^3$. In addition, the distance between the cylinders varied in the experiment. The results showed the the influence of the re-circulation bubble length and the features of vortex formation on the flow pattern and characteristics for the configuration under study in the entire range of regimes. The results were compared with experimental and numerical data from previous literature and found to be in good agreement. Our innovative method based on gradient heatmetry showed high information content and a fairly high accuracy of measurement.

**Keywords:** convective heat transfer; flow; gradient heatmetry; PIV; gradient heat flux sensors; heat transfer coefficient





## 1. Introduction

The flow around a cylindrical surface is of interest to engineers for a number of reasons. Complex flow phenomena, such as boundary layer separation, instability, shear layer interaction, vortex shedding and flow-induced vibration, are associated with cylinder flow features. On the other hand, a strong dependence of forced convective heat transfer on the flow characteristics makes the study of flow around heated cylinders more interesting and challenging. A great number of studies have been performed to gain an understanding of the fluid mechanics and heat transfer of cylinder flow over a wide range of regimes, sizes, shapes and configurations, and a vast amount of literature is available on the subject.

A number of researchers have studied forced convection heat transfer between a cylinder and a fluid flowing around it experimentally (using PIV [1,2], hot-wire anemometry (HWA) [2–4], smoke image velocimetry (SIV) [5], or *Infrared Thermography* [6]) and computationally: by finite-volume method [7], direct numerical simulations (DNS) [8] and large-eddy simulations (LESs) [9–12]. Despite the long history of studying the problem, with the development of science and technology, researchers are trying to understand the physics of the process more deeply, and interest in it is not fading. An example of the last assertion is the paper of Ma and Duan [13]. The authors noticed the complexity and multifactorial nature of flow over regular-shaped bodies. Thence they proposed a new modified drag coefficient which may be a preferable dimensionless parameter to describe fluid flow and heat transfer's physical behavior. It is intriguing that this new method is valid for all Reynolds number regimes.

The first research on this subject was presented in the articles of Kirpichev, Sinelnikov and Gukhman, a review of which is presented in [14]. In these studies, the dependencies of local and averaged heat transfer coefficient (HTC) at the circular cylinder surface depending on the flow regime, the yawed angle and turbulence were obtained. The results of early

studies are summarized perfectly by Zhukauskas [15]. He concludes that heat transfer in a cross-flow around cylinders primarily depends on the Reynolds number. In flow studies, researchers in those years most often used pressure sensors. Further, according to the obtained data, the velocity calculation was performed. This approach is quite simple and informative. A hot-wire anemometer or flow-meter were also used to determine velocity. The advantages are the response time and small size of a hot-wire probe, which does not introduce perturbations into the fluid flow. An improved way to measure hydrodynamic and thermal boundary layers is to apply a thin-film band heater to the surface of the cylinder in addition to HWA. In this way, Wlezien et al. [16] investigated the unsteady flow behind a single cylinder.

From the 1970s to the 2020s, researchers presented many articles and monographs about local and averaged characteristics. For example, Hanson [17] studied the influence of the non-transversality of flow on the formation and frequency of vortex shedding during flow around a single cylinder. Flow around non-circular cylinders has been also investigated. The paper of Norberg [18] reports an experimental investigation of the flow around and pressure forces on rectangular cylinders at various angles of attack. Forces and moments for cylinders for cylinders were estimated using static pressure measurements and wake frequencies and were determined from HWA.

Classical work in this vein is the study by Nakamura and Igarashi [3]. The authors investigated fluctuations by the heat flux sensor HFM-7E by Vatell and infrared thermography. The experimental results of averaged Nusselt numbers coincide with the earlier ones, and in terms of fluctuations, they brought significant novelty. Despite the obvious advantages in measuring the heat flux per unit area at the heat transfer surface, this approach is not widely used. In our opinion, this was due to the weak distribution of the heat flux sensors, some difficulties in their use and limitations which will be discussed below. We were able to find an article by Scholten J. and Murray D. [19] where the heat flux sensor 55R47 by Dantec mounted on the cylinder was used in a similar problem. Heat flux measurements in various angular positions were performed by turning the cylinder around the axis. The authors indicate that a thin-film sensor on a copper cylinder leads to violation of the thermal boundary layer and distortion of the measured Nusselt number. To solve the problem, the authors had to use a complex and laborious correction procedure based on the measured distribution of transverse shear stress. The result of the study was the conclusion that "in the front part of the cylinder the heat flux changes with a frequency equal to the frequency of vortex formation, with the except of front point where fluctuations occur at twice the frequency of vortex formation. For the backward part, the fluctuations are more random with the except for places directly after separation" [19].

There are papers devoted to the study of tandem cylinders. Gu and Sun [20] presented the results of a investigation on the interference between two circular cylinders arranged in a staggered configuration. Their experiments was carried out for a high sub-critical Reynolds number using pressure measurements and the flow visualization. The authors noted the phenomenon of two different pressure patterns at critical angles, which created discontinuity of large lift forces on the cylinders. Other authors [21,22] pointed to the presence of different regimes in the flow around two cylinders depending on the location and regime of the flow, and gave a rather complex classification that takes into account both of these parameters.

Very essential also are the papers dedicated to heat transfer enhancement. The average heat transfer coefficient can be increased, for example, using cooling fins. Real-Ramirez et al. [23] presented a CFD analysis of the hydrodynamics of a finned tube. This paper studies the fluid flow pattern around a tube with angled fins, and special attention is paid to calculating the pressure coefficient for the internal and external surfaces of the inclined fins. As a result, the authors indicated that at approximately $\varphi = 110°$ one horseshoe vortex system forms at the fin's channel. This rather intensive vortex creates significant non-uniformity of the HTC field at the fins' surface. We studied the same vortex and its affect on heat transfer at a single circular fin in our paper [24].

Today, the most considered problem is the hydrodynamics and heat transfer in pulsating flows [25,26]. The relevance is the abundance of the micro-channel devices utilized, among others, in cooling electronics, for which reason the investigated regimes are usual laminar. Most research on the topic is performed through numerical simulations. Over the past decade, there have been hundreds of studies, so we will not list them here. It is only important to note that in almost all papers, the authors point out the lack of experimental data on heat transfer during flow around cylinders of various cross-sections and configurations. In our study, we used them as a data source for setting up verification experiments.

Summarizing the literature data, the following conclusions can be drawn:

- The problem of flow around cylindrical heat transfer surfaces is relevant and still has significance for both scientific research and practice;
- The development of new measurement methods is applied to this subject both for debugging the methodology and for deepening the physical understanding of the processes;
- There is a lack of experimental data for verification of numerical simulations;
- The development of a method for the flow study is ahead of the methods for the heat transfer study, which is caused by the low prevalence of heat flux sensors.

Hence, this study analyzes the influences of Reynolds number and distance between cylinders mounted one after another on flow and heat transfer. An innovative method of measuring heat flows was used in the work—gradient heatmetry. The PIV method was used to correlate the features of the flow and heat transfer.

Our research had two objectives:

(1) To prove the applicability, adequacy and informative value of the proposed method by comparing the main characteristics of the flow and heat transfer obtained by other researchers during the flow around a single heated cylinder with a circular cross-section.
(2) To study the features of flow and heat transfer during flow around a system of two or three cylinders arranged in a row using the proposed method.

The main contributions of the study consisted of the experimental determination of the flow parameters (velocity and its components) and heat transfer parameters (HTC) using the new method, which made it possible to qualitatively and quantitatively evaluate the influences of the geometry of the regime and of the system on these parameters.

## 2. Experimental Methods

### 2.1. Wind Tunnel

The experiments were carried out at the velocity range of 1 to 15 m/s in an open circuit wind tunnel at the Science Educational Center, "Energy Thermophysics," Peter the Great St. Petersburg Polytechnic University. The air flow in the tunnel was generated by a stage centrifugal fan having a thyristor variable-frequency control. A settling chamber provided with honeycomb gauges was used for correcting the flow. The airflow was discharged into the test section through the outlet of the contraction nozzle with an area ratio of 7:1. The test section is provided with an Eiffel chamber made of Plexiglass. In addition, the key feature of this tube is the presence of a heat exchanger which ensures the temperature of the free-stream airflow is almost constant ($\Delta T$ = 0.2 K). The free-stream turbulence intensity in the test section was found to be about 1%. In more detail, specifications of the tube can be found in [14].

### 2.2. Test Models

All the experiments were carried out on hollow circular cylinders 600 mm in length. Models were heated with the saturated steam under atmospheric pressure, thereby keeping the surface temperature constant and close to 100 °C. All models were painted black, so the the heat transfer surface emissivity was 0.99. The surface isothermality was regularly checked with a thermal imager FLIR 650P. Thus, in our experiments, the first-type boundary condition was ensured.

The experiment used two sets of cylinders. The first set was designed to study heat transfer; hollow cylinders of 66 mm diameter and 0.1 mm thickness are made of steel sheets. The model's structure is shown in Figure 1.

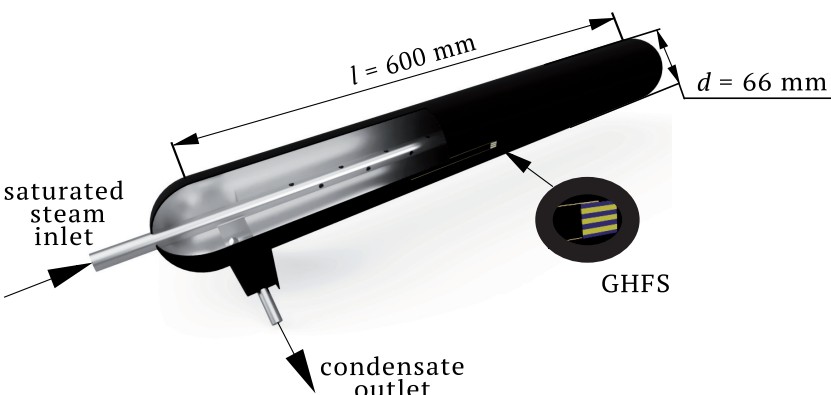

**Figure 1.** Cylinder structure.

A heat flux sensor was installed on the investigated cylinder (the first, second or third), and the cylinder itself was rotated around the axis by an angle $\varphi = 0\ldots180°$. In the case of flow around a row of cylinders, the frame allowed us to move the cylinder and change the distance $S$ (see Figure 2) between the cylinders, which could vary from $0.5d$ to $3d$, where $d$ is the cylinder's diameter. As an illustration, Figure 2 shows a system of two cylinders.

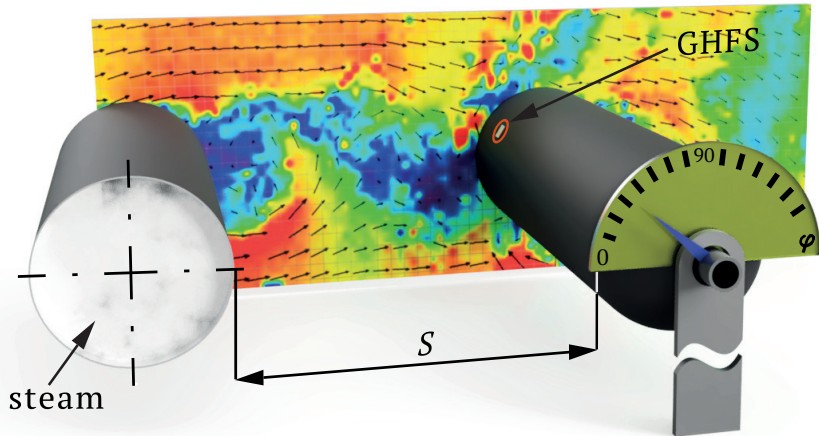

**Figure 2.** Flow around a cylinder pair.

The second model was designed to study flow using PIV, so we used cylinders with a diameter of 25 mm. This is a forced measure, but it made it possible to increase the considered flow area. Since the heat transfer and flow studies were carried out on different models separately and independently, the experiments were carried out at the same Reynolds numbers for the validity of comparing the results.

### 2.3. Gradient Heatmetry

To measure the heat flux per unit area at the surface of the cylinders, a gradient heat flux sensor (GHFS) was used. The GHFS was made by the scientific team of "Energy Thermophysics" and consists of a series of anisotropic thermopower elements (ATEs). An ATE is a material with anisotropic thermophysical and thermopower properties. When affected by heat flux, an ATE generates thermo-EMF $\vec{E}$ normal to heat flux per unit area vector $\vec{q}$ and proportional to the heat flux rate (Figure 3).

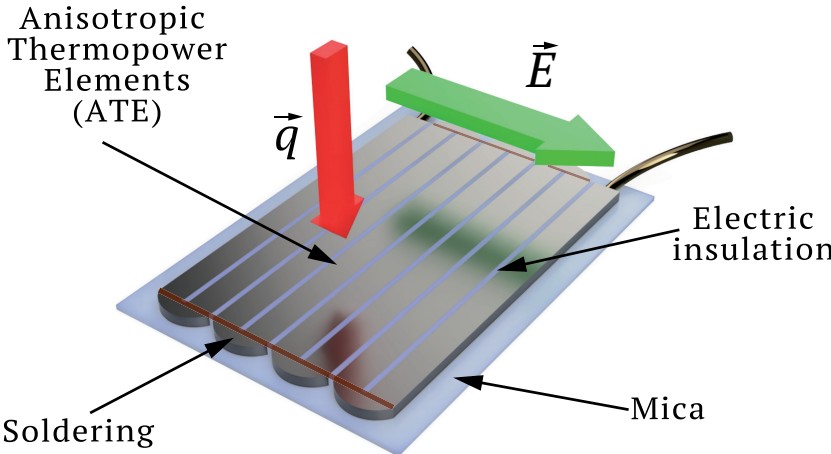

**Figure 3.** Scheme of GHFS.

A more detailed description of the principle of GHFS operation can be found in [27]. It is also worth noting that the sensors have proven themselves in the study of water boiling [28], condensation [29] and forced convection [24].

To calculate the heat flux per unit area, this formula is used:

$$q = \frac{E}{AS_0},$$ (1)

where $E$, mV, is the signal generated by the GHFS; $A$, $m^2$, and $S_0$, mV/W, are GHFS cross area and volt–watt sensitivity, respectively.

We used a battery-type GHFS, the main specifications of which are presented in Table 1.

**Table 1.** Gradient heat flux sensor specifications.

| Material | Sizes, mm | Volt-Watt Sensitivity $S_0$, mV/W | Thermal Stability, K |
|---|---|---|---|
| single-crystal bismuth | $2 \times 2 \times 0.2$ | 7.8 | 544 |

This choice was due to the fact that:

- The temperature on the surface of the model would be close to 100 °C;
- The operating environment is not aggressive;
- The volt–watt sensitivity of bismuth-based GHFS is quite high and practically constant for the specified temperature;
- A double-twisted pair of bismuth bars in the GHFS minimizes the contribution of interference from the operating PIV laser and other pickups;
- Bismuth-based GHFS resistance does not exceed 2...3 Ω, which makes signal conversion and archivation easier.

The GHFS signals were recorded and archived using an NI 9213 analog-to-digital conversion from National Instruments for visualization and primary processing of measured signals in the LabVIEW software.

### 2.4. Particle Image Velocimetry

The velocity measurement was as follows: tracers were introduced into the stream near the streamlined body, which were illuminated twice with a light sheet. Time delay between pulses was selected depending on the flow rate and display scale. Light, reflected and scattered by tracers, was recorded using a high-quality lens on two separate frames of a cross-correlation digital (CCD) camera. The output data were transferred from the camera to the PC's memory. For the pre-assessment, digital PIV record was divided into

small sub-areas called "search areas." The tracer image offset vector at the first and second points in time was determined for each area of the survey using statistical methods. The velocity field in the plane of the light sheet was calculated taking into account the time-delay between two backlights and was refined in system calibration.

In our experiments, we used PIV by POLIS, developed and manufactured at the Institute of Thermal Physics of the Siberian Branch of the Russian Academy of Sciences [30]. It includes a Quantel BSL dual pulse laser, CCD camera, synchronization device, smoke generator and image processing software. The main specifications are tabulated in Table 2.

**Table 2.** PIV system specifications.

| Laser Wavelength, nm | Laser Impulse Energy, MJ | Laser Impulse Time, ns | Camera Resolution, Mp | Maximum of Operation Frequency, Hz |
|:---:|:---:|:---:|:---:|:---:|
| 532 | $2 \times 220$ | 7 | 4 | 8 |

The temporal resolution reaches 10 μs, which made it possible to study the entire the range of Reynolds numbers used in our study. Previously, we had found that for PIV near heated models, the tracers of a standard fog machine for these purposes were not suitable. Weighted oil tracers have time to evaporate over the heated surface, and flow visualization in the boundary layer becomes impossible [14]. We used wood smoke particles obtained with a smoker used by beekeepers.

*2.5. Experimental Procedure*

In various experiments, the scheme of the stand differed only in the model under study (the count of cylinders). In the supplementary description and in the figures, we use a model composed of three cylinders installed one after another. The scheme of the PIV setup is shown in Figure 4.

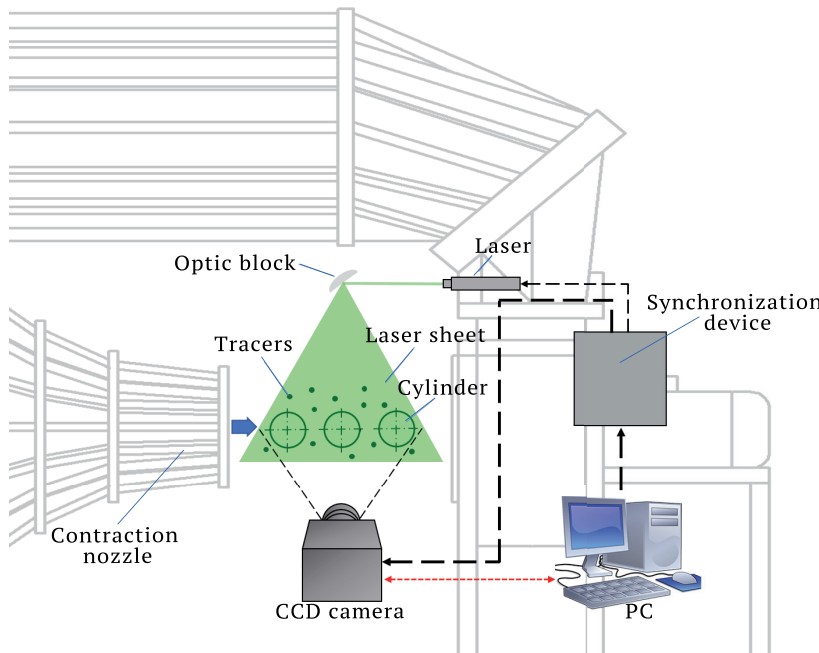

**Figure 4.** Diagram of the experimental PIV setup.

The model was located in an Eiffel chamber of the wind tunnel. The synchronization device of the POLIS PIV-system generates paired laser flashes, and PIV is conducted according to the readings of the CCD camera. At a result, we obtained a flow pattern

(velocity field) near the model. If necessary, the values of the velocity components $u$ and $v$ are displayed in various sections perpendicular to the plane of the laser sheet.

The experimental design for the study of heat transfer is diagrammatically illustrated in Figure 5.

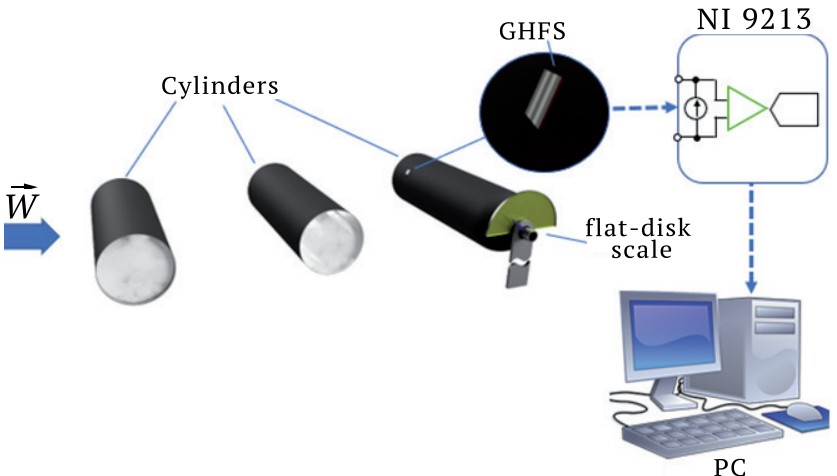

**Figure 5.** Diagram of gradient heatmetry experimental setup.

The GHFS was mounted on the investigated cylinder (the third one in the scheme) equipped with a dial scale to set the angle $\varphi$. All cylinders were heated with saturated steam, and the investigated cylinder rotated around the axis.

The GHFS signals were registered with an ADC and sent to the computer for processing. Experiments used a fixed distance $S$ between the cylinders for all regimes. Afterwards, the distance $S$ changed, and the experiments were repeated.

As a result of heat transfer studies, it was possible to obtain the distributions of the local heat flux and HTC over the surface of the cylinders. Having obtained the distributions of local values, it was possible to find the parameters averaged over the surfaces of the models.

Based on the results obtained by these methods, the following quantities were measured and calculated:

1. Local heat flux per unit area $q_\varphi$, W/m$^2$;
2. Local HTC $h_\varphi$, W/(m$^2$K);
3. Local Nusselt number Nu$_\varphi = \frac{h_\varphi d}{k_f}$ (cylinder diameter $d$, m and thermal conductivity of air $k_f$ were used for the calculation, W/(m K));
4. Surface-averaged Nusselt number Nu $= \frac{hd}{k_f}$ ($h$ is the $h_\varphi$ averaged over $\varphi$ within 0...180°);
5. Absolute velocity $W$ and its projection on the axes $OX$ and $OY$ ($u$ and $v$, respectively) near the surface of the model.

It was necessary to know the free-stream temperatures $T_f$. The temperature was measured in the flow in front of the model using the testo-450i multi-functional device.

*2.6. Measurement Uncertainty*

Uncertainty calculations were carried out according to ISO/IEC GUIDE 98-1:2009—Uncertainty of Measurement [31], according to which uncertainty of measurement is the expression of the statistical dispersion of the values attributed to a measured quantity. Combined standard uncertainty of $y = f(x_1, x_2 \ldots x_i)$ is calculated by the following formula:

$$U_y = \pm\sqrt{\sum (\frac{\partial f}{\partial x_i} U_{x_i})^2}, \tag{2}$$

where $\frac{\partial f}{\partial x_i} U_{x_i}$ is dispersion of $x_i$. The values of $U_{x_i}$ were assumed to be known and were determined by the characteristics of the devices.

The heat flux per unit area on the cylinder surface was measured using 1200 measurements for each angle and on each regime. Factors contributing to uncertainty in the heat flux measurements are:

- GHFS particular error;
- Error in measuring the GHFS's area;
- Measurement error using ADC.

The following is an algorithm for calculating the uncertainty of the local heat flux. In order not to clutter up the formulas, we will not write a subscript. First of all, it is necessary to evaluate type A uncertainty:

$$U_{A_q} = \sqrt{\frac{\sum(q_i - \bar{q})^2}{n(n-1)}}. \tag{3}$$

On the other hand, other errors should be evaluated as a type B uncertainty:

$$U_{B_q} = \sqrt{(\frac{\partial q}{\partial E} U_E)^2 + (\frac{\partial q}{\partial A} U_A)^2 + (\frac{\partial q}{\partial S_0} U_{S_0})^2}, \tag{4}$$

where $q$ be counted according to Formula (1). The relative uncertainty of the ADC $U_E$ is equal to 0.4% according to the testing certificate; combined standard uncertainty of the GHFS area $U_A$ is equal to $2.8 \times 10^{-8}$ m$^2$. Combined standard uncertainty of the GHFS volt–watt sensitivity $U_{S_0}$ was assumed to be equal to $6.02 \times 10^{-2}$ mV/W.

Then, we have combined standard measurement uncertainty for types A and B quantities:

$$U_\Sigma = \sqrt{U_A^2 + U_B^2}. \tag{5}$$

To estimate the extended uncertainty, the coverage factor was $K = 1.2$.

$$U_{\Sigma\,ext} = KU_\Sigma. \tag{6}$$

As a result, we obtained that the relative uncertainty of measuring the heat flux:

$$u_{\Sigma\,ext} = \frac{U_{\Sigma\,ext}}{q} = 1.3\%. \tag{7}$$

The result was obtained for a series of experiments at Re = 9600 at the angle $\varphi = 30°$. The calculation for uncertainty of local HTC measuring was carried out similarly:

$$h = \frac{q}{\Delta T}, \tag{8}$$

so density measurements affect accuracy heat flux and temperature difference. Type B uncertainty was calculated as:

$$U_{B_T} = \sqrt{(\frac{\partial h}{\partial q} U_q)^2 + (\frac{\partial h}{\partial \Delta T} U_{\Delta T})^2}. \tag{9}$$

HTC measurement uncertainty was 2.8%.

Estimating the accuracy of PIV is quite difficult. A detailed overview of sources of uncertainty for PIV is presented in [32]. There are also special soft methods for calculating uncertainty PIV during experiments. We used a short-cut technique. The formula for determining the velocity in its simplest form is written as

$$W = \frac{L}{\tau} k, \tag{10}$$

where $L$ is tracer shift during the time between laser pulses, $\tau$ is the time-frame between laser pulses and $k$ is scale factor. As in the previous case, the standard uncertainty is:

$$U_{B_W} = \sqrt{\left(\frac{\partial W}{\partial L}U_L\right)^2 + (\frac{\partial W}{\partial \tau}U_\tau)^2 + (\frac{\partial W}{\partial k}U_k)^2}, \qquad (11)$$

where $U_L$ is uncertainty of tracer shift measuring and $U_\tau$ is the uncertainty of the time-frame. The uncertainty of measuring scale factor $U_k$, in turn, depends on the number of pixels and the actual distance. For our 4 MP CCD camera, uncertainty of velocity measurements was 7.9%.

## 3. Results

### 3.1. Flow around a Single Cylinder

As a first step, it was necessary to compare our results with the most thoroughly studied flow regime around a single circular cylinder. A special series of experiments was carried out for the Reynolds number Re = 3900. The most important parameter in the problem under consideration is the length of the re-circulation zone (or re-circulation bubble). Re-circulation bubble length $L$ is defined as the distance from the rear stagnation point to the point on the $OX$-axis, where the longitudinal component of the velocity $u$ changes sign from negative to positive.

Figure 6 combines the our PIV results and some experimental [5] and numerical simulation [9,10] data. The curve we obtained (red line) also shows the uncertainty in velocity and position measurements. Good agreement can be noted both with the simulation and with the experiment, especially at a distance of (3...4)$d$ from the rear stagnation point.

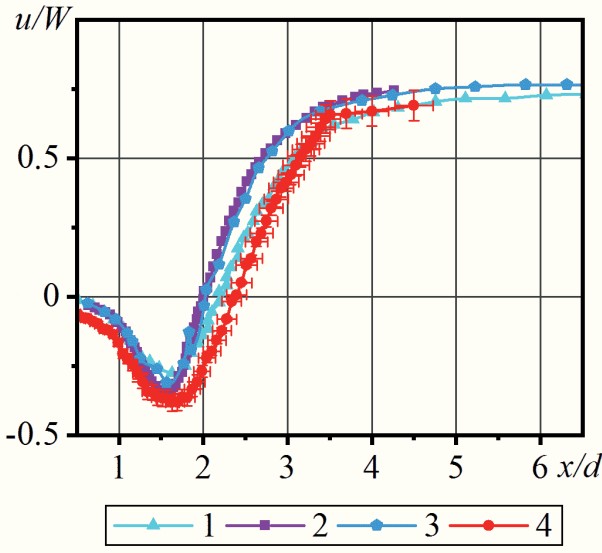

**Figure 6.** Longitudinal velocity component $u$ in the cylinder wake: 1—LES results [10]; 2—SIV results [5]; 3—LES results [9]; 4—our data.

It is seen that, according to our data, our region of negative velocity $u$ was wider than in other works. The estimate $L = 1.79d$ coincides with that obtained in [11] for the temperature difference $\Delta T = 25$ K and corresponds in order of magnitude to the literature data. Heating the cylinder resulted in a reduction in the air density adjacent to the cylinder and a large increase in the viscosity.

In addition, the regime of Re = 3900 was not turbulent, and with a temperature difference of 75 K, the contribution of natural convection was noticeable.

It is also possible to consider the profiles of the time-averaged longitudinal ($u$) and transverse ($v$) velocity components in the cylinder's wake (Figure 7).

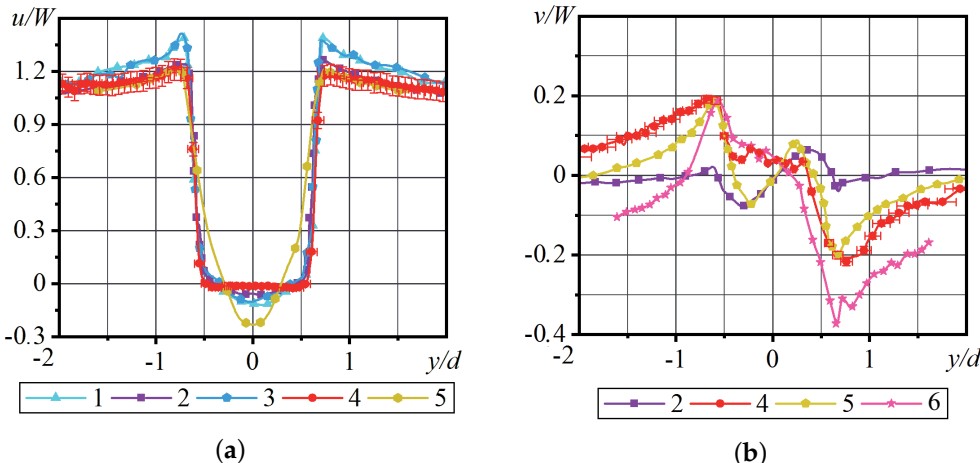

**(a)**                                                                **(b)**

**Figure 7.** Profiles of the longitudinal $u$ (**a**) and transverse $v$ (**b**) velocity components behind the cylinder at $x/d = 1.06$: 1...4 correspond to the designations in Figure 6; 5—DNS results [8]; 6—PIV results [1].

The value $y/d = 0$ corresponds to the cylinder horizontal axis. The profile of the velocity component $u$ changes its pronounced U-shape to a V-shape as it moves away from the cylinder. With $y/d = \pm 1$, an N-shaped change in the transverse flow velocity is observed. For $y/d = 0$, the transverse component is equal to zero throughout the track. The upper and lower vertices are in contact, and consequently, there is no transverse velocity.

Our data qualitatively coincide with the literature results, and the more extreme values can be explained by the heat transfer. Thus, Joggi et al. in [11] indicate that as the temperature drop $\Delta T$ increases, the extrema "move away" from one another.

The next step was to evaluate how the local HTC is distributed over the cylinder surface and how the fluctuation intensity changes at different angles $\varphi$. We used the results of simulations by Jogi et al. [11] and experimental data by Nakamura and Igarashi [3] for the Reynolds number 3000.

Figures 8 and 9 show the changes in the dimensionless local Nusselt number Nu from the angle $\varphi$.

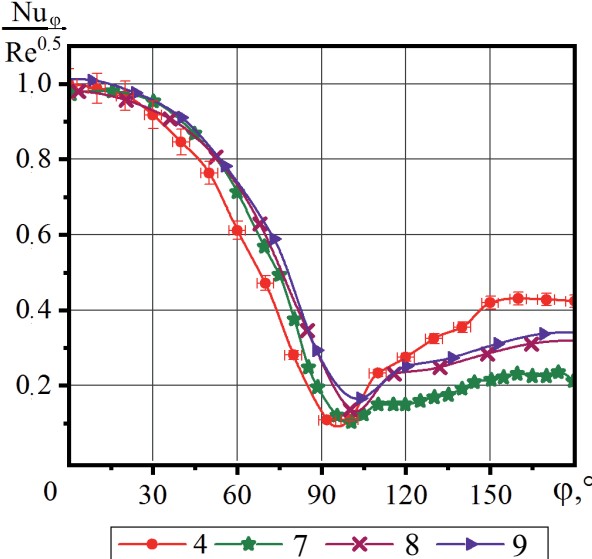

**Figure 8.** Distribution of the local Nusselt number over the cylinder surface: 4—our data; 7—data by Nakamura and Igarashi [3]; 8 and 9—data by Arpan Sircar et al. [12] (fluid properties taken at the flow temperature and at the average temperature of the flow and cylinder wall).

On all curves, the HTC maximum is on the front stagnation point; then HTC decreases to the separation point, after which it slightly increases. In view of the low Reynolds number, the second HTC maximum is absent on the curves. The curve for the HTC reducing to the separation point in our case is steeper, while in the aft part it rises above the others. For the average Nusselt number, the difference is compensated and the dimensionless HTC averaged over the surface differs by 2...3% of those accepted in the literature. This is within the experiment uncertainty. According to our data, the boundary layer separation occurs a little earlier. Figure 9 shows the curves obtained for the same conditions, but for different temperature drops. As $\Delta T$ increases, the separation point shifts upstream.

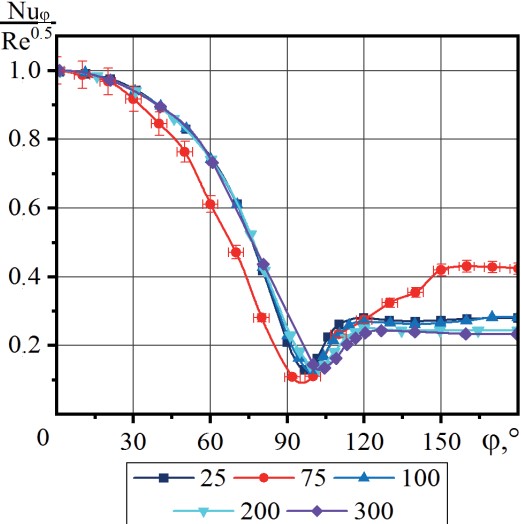

**Figure 9.** Distribution of the local Nusselt number over the cylinder surface: the legend shows temperature differences $\Delta T$, K (75—our data, other—data by Jogee et al. [11]).

In general, the results of gradient heatmetry and PIV are in good agreement with the other researchers' data. This made it possible to consider the new technique workable and apply it to other models.

### 3.2. Flow around Two Cylinders

By analogy with a single cylinder, we show the distributions of the dimensionless HTC on the second cylinder's surface. Dependencies of the Nusselt number from the angle $\varphi$ are supplemented by velocity fields near the cylinders obtained by PIV. Figure 10 shows the HTC curves for a fixed distance between the cylinders $S$ and for various regimes.

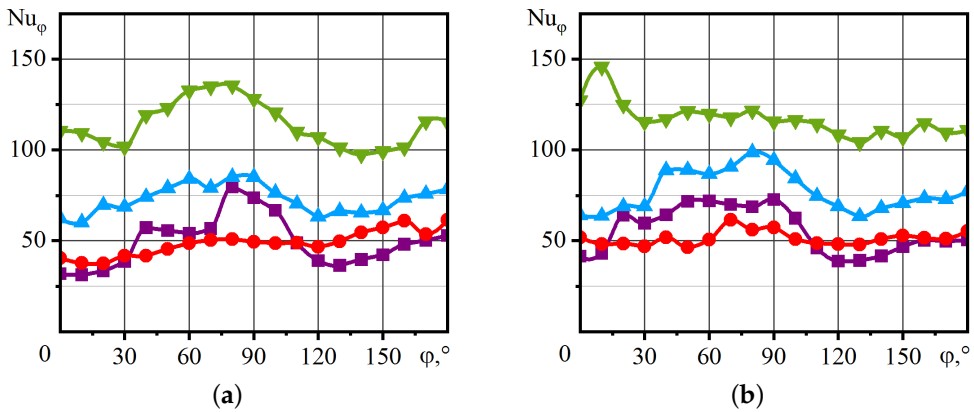

**Figure 10.** Distribution of the local Nusselt number over the second cylinder: (**a**) $S = d$ and (**b**) $S = 2d$ (purple curve—Re $= 2.4 \times 10^3$; red curve—Re $= 4.8 \times 10^3$; blue curve—Re $= 9.6 \times 10^3$; green curve—Re $= 29.8 \times 10^3$).

One can see noticeable differences between the HTC curves and those obtained for a single cylinder. The maximum is shifted from the front stagnation point, and its position depends both on the Reynolds number and on the distance $S$. There is also no explicit minimum. For low flow rates, the curve is strongly nonmonotonic. Just starting around with Re = $10^4$, one can observe the similarity of the curves.

The course of the curves can be explained by considering the velocity fields near the second cylinder. The local Nusselt number on the surface of the second cylinder depends on which part of the wake it is located in. For example, an HTC increase in the range of angles $\varphi$ = 40...100° for the case $S = d$ can be observed for all regimes. There is a maximum deviation of the transverse velocity component $v$ at this distance from the first cylinder. Further, the vortex is destroyed, and the turbulent stream flows around the aft part of the second cylinder approximately uniformly. As a result, the HTC curves in the last third of the graph are almost equidistant.

As the distance $S$ increases to $2d$, the described mechanism of is retained for low Re numbers. The curve for Re = $20.2 \times 10^3$ has the form of a damped harmonic oscillation, and the curve for Re = $28.8 \times 10^3$ practically becomes straight, except for the peak near the front stagnation point. It can be explained that with an increase in the flow velocity the length of the re-circulation bubble decreases and the effect of a large-scale vortex on heat transfer weakens on the second cylinder.

There is a difference in the wake behind the second cylinder and that behind a single cylinder. The length and width of the re-circulation zone are smaller, and its dimensions are more sensitive to the regime. For ease of comparison, Figure 11 shows longitudinal velocity $u$ in the wake at different distances from the second cylinder.

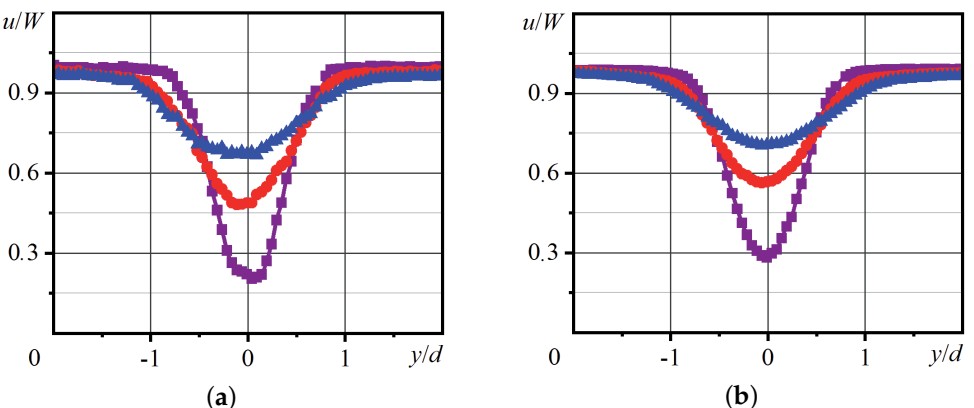

**Figure 11.** Profiles of the longitudinal velocity component $u$ at: (**a**) Re = $4.8 \times 10^3$ and (**b**) Re = $20.2 \times 10^3$ ($S = 2d$: purple curve—$x/d = 1.06$; red curve—$x/d = 1.54$; blue curve—$x/d = 2$).

In contrast to the wake behind the first cylinder, the longitudinal velocity profile behind the second cylinder has a V shape. The section where the profile has a U-shape is absent, which indicates the absence of flow restructuring.

Figure 12 shows the transverse velocity component $v$ in the same sections as in Figure 11.

It can be seen that the flow regime has little effect on the courses of the curves. All curves are similar to the curves obtained for a single cylinder; however, the zone where the transverse component differs from zero in the longitudinal direction (along the $y$-axis) is smaller. Changes along the $OY$ axis behind the second cylinder propagate to a greater distance.

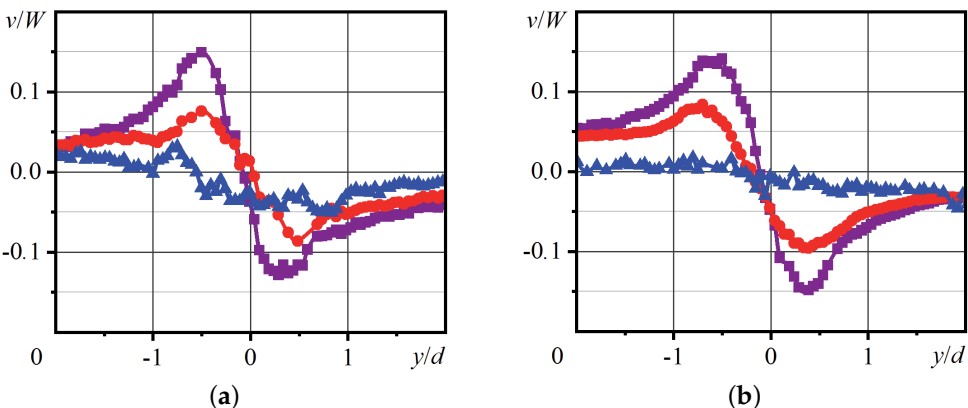

**Figure 12.** Profiles of the transverse velocity component $v$ at: (**a**) Re = 4.8 × $10^3$ and (**b**) Re = 20.2 × $10^3$ ($S = 2d$: purple curve—$x/d$ = 1.06; red curve—$x/d$ = 1.54; blue curve—$x/d$ = 2).

### 3.3. Flow around Three Cylinders

As before, the dependencies of local HTC on angle $\varphi$ for the third cylinder are shown. Graphs Nu = $f(\varphi)$ are supplemented with velocity fields near the third cylinder. The HTC distributions over the surface of the third cylinder for different regimes and at a fixed distance $S$ obtained by gradient heatmetry are shown in Figure 13.

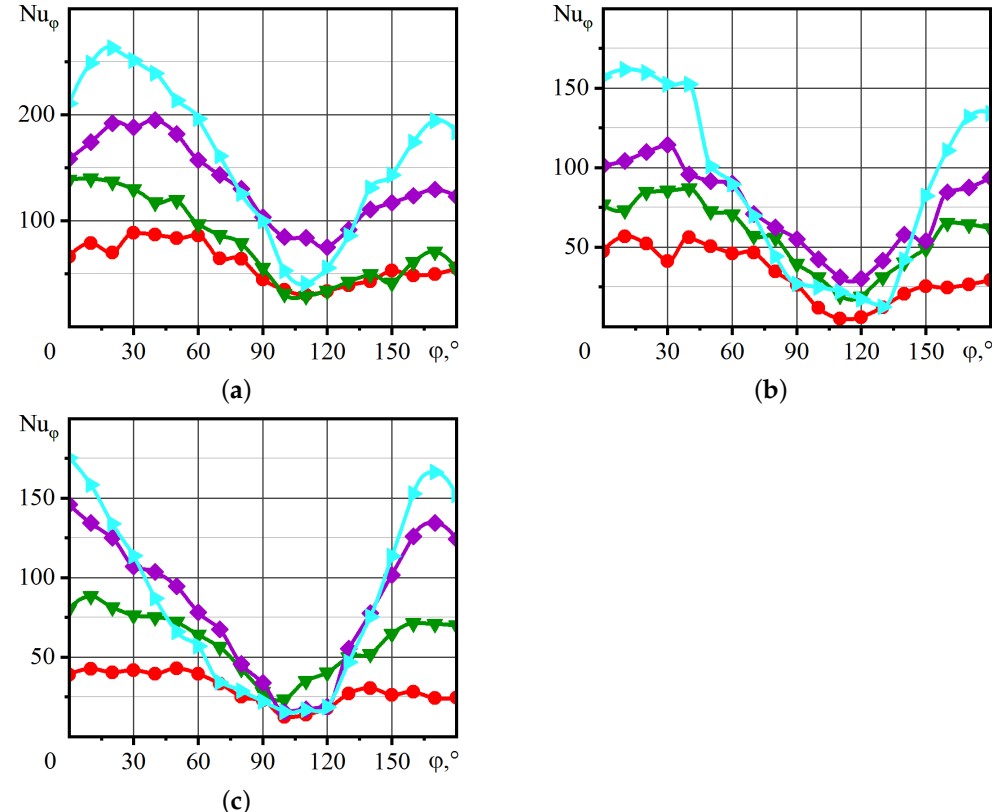

**Figure 13.** Distribution of the local Nusselt number over the third cylinder: (**a**) $S = d$, (**b**) $S = 2d$ and (**c**) $S = 3d$ (red curve—Re = 2.4 × $10^3$; green curve—Re = 9.6 × $10^3$; purple curve—Re = 20.2 × $10^3$; turquoise curve—Re = 29.8 × $10^3$).

The HTC distributions near the third cylinder resemble those for a single cylinder but with slight differences. HTC maximum is not located on the frontal section. Additionally, the curve runs quietly. The HTC minimum is for the angle $\varphi$ = 100...110°, after which the curve's behavior depends on the regime. The influence of the distance between the cylinders is noticeable. At $S = 1d$, the HTC is approximately 1.5 times higher. It was

found that for some regimes, the surface-averaged Nusselt number can exceed that of a single cylinder.

When considering the velocity field, it is seen that the length of the wake behind the third cylinder weakly depends on the parameter $S$. Profiles of the longitudinal and transverse velocity components in the wake behind the third cylinder are shown in Figure 14.

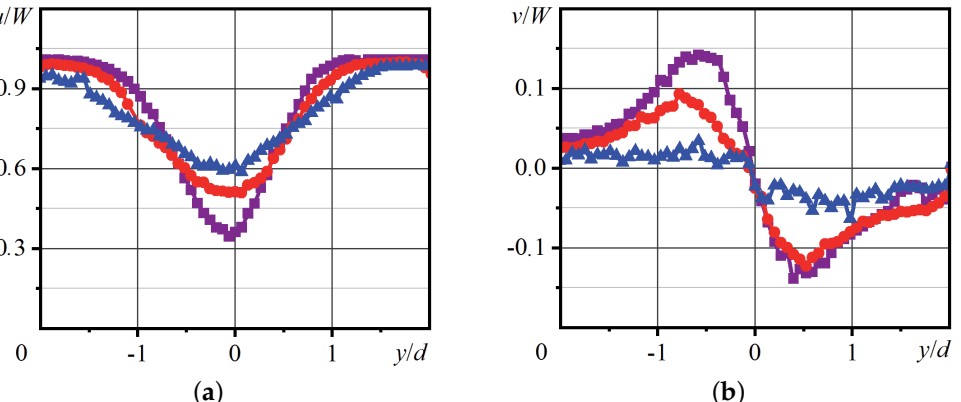

**Figure 14.** Distribution of longitudinal (**a**) and transverse (**b**) velocity in the wake behind the third cylinder at Re = $9.6 \times 10^3$: purple curve—$x/d$ = 1.06; red curve—$x/d$ = 1.54; blue curve—$x/d$ = 2.

On the other hand, the width of the wake, where the longitudinal velocity is lower than the free-stream velocity, is greater than that behind the first or second cylinder.

## 4. Discussion

The paper described a unique multi-method including gradient heatmetry and particle image velocimetry. The main goal was testing the multi-method, comparing data with the literature and estimating its advantages and disadvantages. The first part of the paper presented techniques used in the work:

1. Particle image velocimetry used to visualize velocity field and obtain velocity profiles $u$ and $v$;
2. Gradient heatmetry used to obtain local heat flux per unit area $q_\varphi$;
3. Temperature measurement used to calculate heat transfer coefficient $h_\varphi$.

The independent variable was the Reynolds number Re = $\frac{Wd}{v}$, where $W$, m/s, is free-stream velocity; $d$, m is the cylinder diameter; and $v$, m$^2$/s, is viscosity at free-stream temperature $T_f$.

The experimental setup and models used in the work which are hollow cylinders heated by saturated steam are described. We used the following configurations:

1. Single cylinder;
2. Row arranged pair of cylinders;
3. Row arranged three cylinder.

The uncertainty of measurements was calculated according to ISO/IEC GUIDE 98–4:2012—Uncertainty of Measurement [31]. The calculations showed that during the measurements, the following results were obtained. The measurement capabilities of the proposed method can be considered satisfactory.

The next step was devoted to a comparison of the cross-flow around a single circular cylinder for Re = 3900, as for the most thoroughly studied model. To describe the flow and heat transfer, the focuses were the velocity fields and their projections on the $x$ and $y$ axes in the wake behind the cylinder ($u$ and $v$, respectively), the length of the re-circulation zone $L$, the separation angle $\varphi_{sep}$ and the separation frequency (Sh) of vertices. The comparison of velocity fields and heat flux distribution was considered in detail. In general, the results of the overwhelming majority performed under similar conditions agree satisfactorily. Our

results are also within the specified ranges. For convenience, the results are presented in Table 3.

**Table 3.** Flow parameters comparison.

| Author and Method | Separation Angle $\varphi_{sep}$,° | Re-Circulation Zone Length $L/d$ | Strouhal Number Sh |
|---|---|---|---|
| Parnaudeau et al. [2]; PIV + HWA | – | 1.51 | $0.208 \pm 0.002$ |
| Parnaudeau et al. [2]; LES | 88 | 1.56 | $0.208 \pm 0.001$ |
| Lourenco et al. [1]; PIV | 86 | 1.19 | 0.22 |
| Kravchenko et al. [33]; LES | 82 | 1.37 | 0.21 |
| Ma et al. [34]; LES | – | 1.59 | 0.219 |
| Norberg [35]; LDV | 80 | 1.66 and 1.40 | 0.21 |
| Dong et al. [36]; PIV | – | 1,47 | – |
| Dong et al. [36]; DNS | – | 1.27 | 0.2152 |
| Abrahamsen Prsic et al. [37]; LES | – | 1.27 | 0.2152 |
| Our results; GH + PIV | 87 | 1.79 | 0.21 |

The differences can be explained by the presence of heat transfer. The separation point shifts; that is, the process is similar to the flow around a cylinder at a higher Reynolds number. As a consequence, the flow in the cylinder wake also changes, which is associated with an increase in the viscosity with increasing temperature.

The work was completed by studying the flow around a system of two and three cylinders. The local HTC distributions over the cylinder surface were obtained for various regimes and distances $S$ between the cylinders. A special effect of $S$ on the heat transfer of the second cylinder was revealed, since it interacts with an intense vortex that has separated from the first one.

In a practical application, it is more interesting to consider the surface-averaged Nusselt number. Figure 15 shows the experimental points of curve Nu(Re) for the second HTC cylinder.

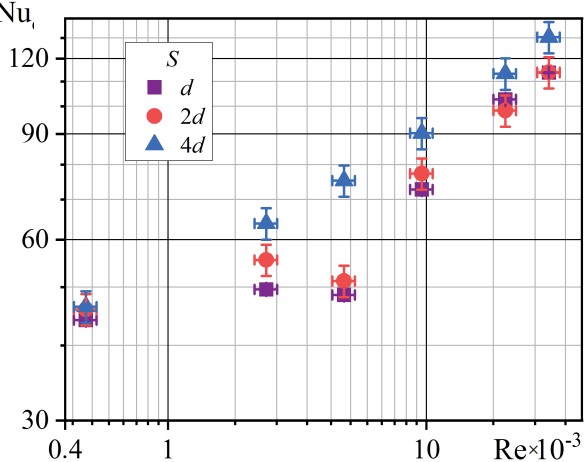

**Figure 15.** Dependence of the surface-averaged Nusselt number for the second cylinder from the regime.

The surface-averaged HTC for the second cylinder was lower than for the first one. The points obtained for the maximum distance *S* lay closest to the straight.

The HTC distributions over the third cylinder resemble the same dependencies as for a single cylinder with a slight difference. The HTC maximum is near the front stagnation point, and the curve is flat. The HTC minimum is also shifted closer to the rear point. At low velocities (Re < $10^3$), HTC slightly increases up to the rear stagnation point. For high velocities, there is an intensive HTC increase after the minimum. The distance between the cylinders *S* noticeably affects the HTC. At *S* = *d*, the HTC was the largest in all regimes, and it was greater than the HTC for a single cylinder. However, for *S* > 2*d*, the average HTC fell below the first one. The length of the wake behind the third cylinder weakly depends on the parameter *S*, but the width of the re-circulation bubble is larger than that behind a single or second cylinder. A compilation of the results obtained for the third cylinder is shown in Figure 16.

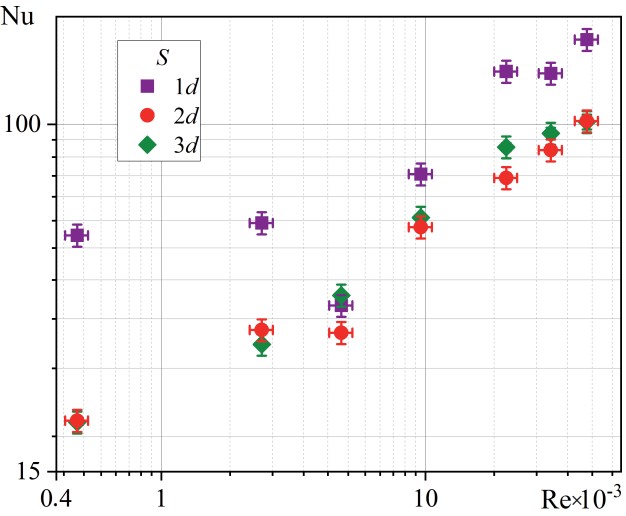

**Figure 16.** Dependence of the surface-averaged Nusselt number for the third cylinder from regime.

According to our data, the distance *S* = *d* is the most effective for heat transfer on the third cylinder. For all Re and *S*, the average Nusselt number over the third cylinder is greater than for the second.

### 5. Conclusions

In the present work, the technique combining gradient heatmetry and PIV was applied in the study of flow and heat transfer. The convective heat transfer from a row-arranged system of cylinders in cross-flow was experimentally investigated for a range of Reynolds numbers (480 < Re 48,000) and distances (*d* < *S* < 4*d*). Experiments were carried out in an wind tunnel. Gradient heat flux sensors were used to calculate heat transfer coefficient, and PIV was employed to measure velocity and velocity components. An IR thermal camera was used to obtain cylinder surfaces temperature, and a multi-functional device was used to measure free-stream temperature. Both regime and system geometry were found to affect the HTC distribution on the cylinders and velocity profile behind. Perhaps, on the basis of new data, it will be possible to improve the methods of designing heat exchangers, modernize existing equipment and create new, more efficient equipment.

The main conclusions are:

- It was shown that the distributions of HTC and the flow velocity components derived from the gradient heatemtry and PIV were in sufficient agreement with data experimental (by SIV, HWA, LDA) and numerical (by DES and LES simulation) data.
- The heat transfer was found to affect the re-circulation zone and velocity profile behind a single cylinder. The heat transfer had a great impact on flow for the transient regime

(Re = 3900). Heating the cylinder resulted in a decrease in the air density and an increase in the viscosity.

- During flow around two or three cylinders installed in a row, the flow and heat transfer are affected by both the regime and the distance *S*. As the Reynolds number increases, the effect of distance *S* weakens.
- The HTC distributions over the surface of the first and third cylinders qualitatively coincide, and the distribution over the second cylinder differs. The second cylinder's HTC distribution is more sensitive to the distance *S* compared to the third cylinder's local HTC.
- The surface-averaged HTC for the second cylinder is less than those for the first and third cylinders throughout the entire studied regimes and distances *S*. At a Reynolds number of 4800, a decrease in the surface-averaged HTC was found.
- The combination of methods makes it possible to study the flow and heat transfer in a single experiment and get more information about the convective heat transfer.

In the future, it is planned to develop the area of research, namely:

- Investigate flows and heat transfer in the same regimes for a system of staggered-arranged cylinders.
- Determine the drug coefficients for the entire system and each cylinder separately.
- Investigate both configurations (staggered- and row-arranged) for oval and airfoil-shaped cylinders.
- Enhance heat transfer with acceptable hydraulic parameters on the described cylinder models using active and passive methods of heat transfer augmentation.

**Author Contributions:** Conceptualization, V.S. and A.G.; methodology, A.G.; validation, V.S.; formal analysis, A.G.; investigation, V.S. and A.G.; data curation, A.G.; writing—original draft preparation, V.S.; writing—review and editing, V.S.; visualization, V.S.; supervision, A.G.; project administration, A.G.; funding acquisition, V.S. and A.G. All authors have read and agreed to the published version of the manuscript.

**Funding:** This research was funded by Russian Science Foundation grant number 22-29-00156, https://rscf.ru/project/22-29-00156/ (accessed on 1 July 2022).

**Institutional Review Board Statement:** Not applicable.

**Informed Consent Statement:** Not applicable.

**Data Availability Statement:** Not applicable.

**Conflicts of Interest:** The authors declare no conflict of interest.

## Abbreviations

The following abbreviations are used in this manuscript:

| | |
|---|---|
| HFS | Heat Flux Sensor |
| EMF | Electromotive Force |
| ATE | Anisotropic Thermoelemnt |
| GHFS | Gradient Heat Flux Sensor |
| GH | Gradient Heatmetry |
| PIV | Particle Image Velocimetry |
| CCD | Cross-Correlation Digital (camera) |
| HTC | Heat Transfer Coefficient |
| ADC | Analog to Digital Converter |
| HWA | Hot-Wire Anemometry |
| LES | Large-Eddy Simulation |
| DNS | Direct Numerical Simulation |
| CFD | Computational Fluid Dynamics |
| LDA | Laser Doppler Anemometry |
| LDV | Laser Doppler Velocimetry |

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
