# Peer review of "Gradient Heatmetry and PIV Investigation of Heat Transfer and Flow near Circular Cylinders"

_inventions, doi:10.3390/inventions7030080_

Round 1

Reviewer 1 Report

This manuscript combined gradient heatmetry and PIV to study the flow and heat transfer in a system of row-arranged cylinders, which is beneficial for improving the methods of designing heat exchangers, equipment, and more efficient equipment. Besides, some issues should be addressed.

1. Section 2.2, what’s the Dirichlet boundary condition?

2. The authors conducted experiments with the same Reynolds numbers. What’s its specific value of it?

3. The authors mentioned that “The temporal resolution reaches 10 μs, which made it possible to study the entire range of Reynolds numbers used in our study”. What’s the main reason?

4. The concrete meaning of variables “μ” and “ν” in the text near Fig. 5 is unclear.

5. At the end of Section 2, how did the authors calculate the uncertainty?

6. Fig. 6 shows the difference in velocity between the three studies and this study. What’s the main reason that caused the phenomenon?

7. Some typical studies on thermal properties should also be cited (Li, K. Q., Li, D. Q., & Liu, Y. (2020). Meso-scale investigations on the effective thermal conductivity of multi-phase materials using the finite element method. International Journal of Heat and Mass Transfer, 151, 119383; Li KQ, Miao Z, Li DQ, Liu Y. (2022). Effect of mesoscale internal structure on effective thermal conductivity of anisotropic geomaterials. Acta Geotechnica, 17, 3553-3566. )

Comments for author:

This manuscript combined gradient heatmetry and PIV to study the flow and heat transfer in a system of row-arranged cylinders, which is beneficial for improving the methods of designing heat exchangers, equipment, and more efficient equipment. Besides, some issues should be addressed.

  1. Section 2.2, what’s the Dirichlet boundary condition?
  2. The authors conducted experiments with the same Reynolds numbers. What’s its specific value of it?
  3. The authors mentioned that “The temporal resolution reaches 10 μs, which made it possible to study the entire range of Reynolds numbers used in our study”. What’s the main reason?
  4. The concrete meaning of variables “μ” and “ν” in the text near Fig. 5 is unclear.
  5. At the end of Section 2, how did the authors calculate the uncertainty?
  6. 6 shows the difference in velocity between the three studies and this study. What’s the main reason that caused the phenomenon?
  7. Some typical studies on thermal properties should also be cited (Li, K. Q., Li, D. Q., & Liu, Y. (2020). Meso-scale investigations on the effective thermal conductivity of multi-phase materials using the finite element method. International Journal of Heat and Mass Transfer, 151, 119383; Li KQ, Miao Z, Li DQ, Liu Y. (2022). Effect of mesoscale internal structure on effective thermal conductivity of anisotropic geomaterials. Acta Geotechnica, 17, 3553-3566. )

Author Response

Dear Reviewer,

Thank you very much for your review and remarks.

In essence, the remarks made can be answered as follows:

Point 1. Section 2.2, what’s the Dirichlet boundary condition?

Response 1: Under the Derichlet boundary conditions we mean the boundary conditions of the first type, that is, we set the surface temperature distribution.

Point 2. The authors conducted experiments with the same Reynolds numbers. What’s its specific value of it?

Response 2: The comment about same Reynolds numbers appears in the text, since the experiments on the study of heat transfer and flow were carried out separately and independently. We have researched Re = 480, 2400, 3900, 4800, 9600, 20200, 29800 and 39400.

Point 3. The authors mentioned that “The temporal resolution reaches 10 μs, which made it possible to study the entire range of Reynolds numbers used in our study”. What’s the main reason?

Response 3: PIV reconstructs the velocity field from two photo of the air-flow with tracers taken with a time difference of τ. For accurate results, the movement of particles between these frames should be small and constant for different velocities. When increasing the flow-rate, it is necessary to reduce the time difference τ or change the computational domain. Our CCD camera with the ability to take two pictures with a difference of 10 µs allowed us to correctly determine the velocity field for high flow rates (untill Re = 4×104 for our model).

Point 4. The concrete meaning of variables “μ” and “ν” in the text near Fig. 5 is unclear.

Response 4: In our study, the letter u denotes longitudinal velocity component and v is transverse velocity component. The decoding of the designations u and v was below in the text. Now moved at first mention. 

Point 5. At the end of Section 2, how did the authors calculate the uncertainty?

Response 5: A detailed uncertainty calculation is added in Section 2.6. "Measurement uncertainty".

Point 6. Fig. 6 shows the difference in velocity between the three studies and this study. What’s the main reason that caused the phenomenon?

Response 6: The curves in Figure 6 taken from the other authors studies were obtained for an isothermal flow around single cylinder, while in our experiments there is heat transfer. The curve is designed to determine the length of the re-circulation zone, and the graphs show that heat transfer affects the flow structure in the wake of a single cylinder. These conclusions are added to the description of the graph and analysis of the curve behavior.

Point 7. Some typical studies on thermal properties should also be cited (Li, K. Q., Li, D. Q., & Liu, Y. (2020). Meso-scale investigations on the effective thermal conductivity of multi-phase materials using the finite element method. International Journal of Heat and Mass Transfer, 151, 119383; Li KQ, Miao Z, Li DQ, Liu Y. (2022). Effect of mesoscale internal structure on effective thermal conductivity of anisotropic geomaterials. Acta Geotechnica, 17, 3553-3566. )

Response 7: We would like to thank you for this comment. These studies are not related to the subject of this investigation since we don't consider the geomaterials properties. However, we have read and familiarized with the proposed articles and it is possible that we will try to use our sensors for the experimental determination of thermal conductivity including thermal conductivity of multi-phase materials.

Reviewer 2 Report

First, congratulations on writing such a valuable article.

However, I propose to improve the article, in particular, the analysis of the literature taking into account the items from the MDPI publishing house.

1.        The structure of the paper is good. First, the introduction presents the theme briefly. Then, the paper has a literature review section and a problem description section.

2.        The objective and contribution could be written more clearly in the introduction section.

3.        The literature analysis presented in the paper can be improved. It was prepared briefly and does not take into account all the aspects relevant to the scope of research included in the article.

4.        It is unacceptable to group the items of literature in such a large number, eg: [1-6]. Line 24.

5.        It is unacceptable to group the items of literature in such a large number, eg: [7-12]. Line 25.

6.        It is unacceptable to group the items of literature in such a large number, eg: [16-20]. Line 43.

7.        Lines 96

The correct notation is: 100ËšC

8.         The conclusions were not supported by the research results.

9.        I propose to include articles from the MDPI publishing house in the literature analysis.

Author Response

Dear Reviewer,

Thank you very much for your review and valuable comments. Following remarks:

Point 1. The structure of the paper is good. First, the introduction presents the theme briefly. Then, the paper has a literature review section and a problem description section.

Response 1: Thanks for the comment.

Point 2.  The objective and contribution could be written more clearly in the introduction section.

Response 2: At the end of section "Introduction" the study objective and contribution are formulated.   Our research has two objectives:

1) To prove the applicability, adequacy and informative value of the proposed method, by comparing the main characteristics of the flow and heat transfer obtained by other researchers during the flow around a single heated cylinder of circular cross-section.

2) To study the features of flow and heat transfer when flowing around a system of two and three cylinders arranged in a row using the proposed method.

The main contribution of the study consits in the experimental determination of the flow parameters (velocity and its components) and heat transfer parameters (heat transfer coefficient) using a new method, which made it possible to qualitatively and quantitatively evaluate the regime and system geometry influence on these parameters.

Point 3. The literature analysis presented in the paper can be improved. It was prepared briefly and does not take into account all the aspects relevant to the scope of research included in the article.

Response 3: The literature analysis presented in the Introduction has been expanded. 

Point 4. It is unacceptable to group the items of literature in such a large number, eg: [1-6]. Line 24.

Response 4: The items of literature sources have been ungrouped and explained

Point 5.  It is unacceptable to group the items of literature in such a large number, eg: [7-12]. Line 25.

Response 5: The items of literature sources have been ungrouped and explained

Point 6. It is unacceptable to group the items of literature in such a large number, eg: [16-20]. Line 43.

Response 6: The items of literature sources have been ungrouped and explained.

Point 7.  Lines 96  The correct notation is: 100 ËšC.

Response 7:  Annoying misprint corrcted

Point 8.  The conclusions were not supported by the research results.

Response 8:  Section "Conclusion" rewritten.

Point 9.  I propose to include articles from the MDPI publishing house in the literature analysis.

Response 9:  Articles from the MDPI publishing house in included the literature analysis:

Real-Ramirez, C.A.; Carvajal-Mariscal, I.; Gonzalez-Trejo, J.; Gabbasov, R.; Miranda-Tello, J.R.; Klapp, J. Numerical Simulations of the Flow Dynamics in a Tube with Inclined Fins Using Open-Source Software. Fluids 2022, 7, 282.

Haibullina, A.; Khairullin, A.; Balzamov, D.; Ilyin, V.; Bronskaya, V.; Khairullina, L. Local Heat Transfer Dynamics in the In-Line Tube Bundle under Asymmetrical Pulsating Flow. Energies 2022

Ali, U.; Islam, M.; Janajreh, I.; Fatt, Y.; Alam, M.M. Flow-Induced Vibrations of Single and Multiple Heated Circular Cylinders: A Review. Energies 2021, 14, 8496

Ma, H.; Duan, Z. Similarities of Flow and Heat Transfer around a Circular Cylinder. Symmetry 2020, 12, 658.

Reviewer 3 Report

The paper presents the experimental results of applying the combined gradient thermal
measurement and PIV technology to a 1-3 row cylinder system. The comments are as follows

1. There are many expression errors and typos including followings
- Line 28~29; The number of authors of the reference [13] is not correct.
- Line 28~29, Line 32, etc; When citing references, describe only the authors last name.
- Line 362; "reveal they correlation" should be corrected.
- Line 365 ~ 368; They are grammatically incorrect and need to be corrected.
2. Uncertain analysis is not clear and should be reported systematically.
3. Present the GHFS and PIV specifications in detail (in tabular forms).
4. In conclusion section. it is necessary briefly summarize the new findings from the study and future research direction.

Author Response

Dear Reviewer,

Thank you very much for your review and comments. The following corrections have been made based on your feedback.

Piont 1. There are many expression errors and typos including followings
- Line 28~29; The number of authors of the reference [13] is not correct.
- Line 28~29, Line 32, etc; When citing references, describe only the authors last name.
- Line 362; "reveal they correlation" should be corrected.
- Line 365 ~ 368; They are grammatically incorrect and need to be corrected.

Response 1:  The text is checked for typos, grammatical and punctuation errors.

Piont 2. Uncertain analysis is not clear and should be reported systematically.

Response 2: A detailed uncertainty calculation is added in Section 2.6. "Measurement uncertainty".

Piont 3. Present the GHFS and PIV specifications in detail (in tabular forms).

Response 3: The main characteristics of the GHFS and PIV used in the study are presented in the form of tables 1 and 2.

Piont 4. In conclusion section. it is necessary briefly summarize the new findings from the study and future research direction.

Response 4: Section "Conclusion" rewritten according to the comments of the reviewers.

Round 2

Reviewer 2 Report

The authors addressed all my comment properly.

Author Response

Dear Reviewer,

Thank you for positive review !

Reviewer 3 Report

The paper has been revised according to the comments.  The paper can be published after adding the following information.

  • Authors used IR-Camera to measure the cylinder surface temperature, and so it will be better to add information about the surface emissivity.  

Author Response

Dear Reviewer,

Thank you very much for positve review. 

All experimental models are painted black, so the the heat transfer surface emissivity is 0.99. This sentence has been added to the experimental models description.